# An Open Source Validation System for Continuous Arterial Blood Pressure Measuring Sensors

**DOI:** 10.3390/s25165173

**Published:** 2025-08-20

**Authors:** Attila Répai, Sándor Földi, Péter Sótonyi, György Cserey

**Affiliations:** 1Jedlik Innovation Ltd., Práter u. 50/A, 1083 Budapest, Hungary; 2Faculty of Information Technology and Bionics, Pázmány Péter Catholic University, Práter u. 50/A, 1083 Budapest, Hungary; foldi.sandor@itk.ppke.hu (S.F.); cserey@itk.ppke.hu (G.C.); 3Faculty of Medicine, Department of Vascular and Endovascular Surgery, Semmelweis University, Városmajor u. 68, 1122 Budapest, Hungary

**Keywords:** radial pulse measurement, blood pressure waveform, simulator, sensor validation, cam

## Abstract

**Highlights:**

**What are the main findings?**
An electromechanical, cam-based continuous arterial blood pressure simulator consisting of cost-effective 3D printable parts was developed. This simulator is capable of accurately repeating waveform samples consisting of multiple cardiac cycles.An open-source Python software package is provided to generate new cams from custom waveforms and validate sensors against them.

**What is the implication of the main finding?**
Hardware components of the proposed simulator are easy to fabricate even within academic or clinical institutions due to the widespread availability of low-cost 3D printing technology. Thus, early-stage development of continuous non-invasive blood pressure sensors benefits from an accessible and easily customizable tool.The validation software provides an extensible framework to store, organize, and evaluate sensor data and to generate statistical reports from them.

**Abstract:**

The advancement of sensor technologies enables the measurement of high-quality continuous blood pressure signals, which has become an important area in healthcare. The development of such application-specific sensors can be time-consuming, expensive, and difficult to test or validate with known and consistent waveforms. In this manuscript, an open-source blood pressure waveform simulator with a Python validation package is described. The core part, a 3D-printed cam, can be generated based on real blood pressure waveforms. The validation software framework compares in detail the waveform used to design the cam with the time series from the sensor being validated. The simulator was validated using a 3D force sensor. The RMSE of accuracy was 1.94 (44)–2.74 (63)%, and the Pearson correlation with the nominal signal was 99.84 (13)–99.39 (18)%. As for precision, the RMSE of the repeatability of cam rotations was 1.53 (71)–2.13 (116)% and the Pearson correlation was 99.85 (16)–99.59 (57)%. The presented simulator proved to be robust and accurate in short- and long-term use, as it produced the signal waveform reliably and with high fidelity. It reduces development costs for early-stage sensor development and research, offering a solution that is easy to manufacture yet capable of continuously outputting human arterial blood pressure waveforms spanning multiple consecutive cardiac cycles.

## 1. Introduction

Measuring the continuous noninvasive blood pressure (CNIBP) waveform is becoming increasingly important and is being studied more frequently. The main reasons behind it are the prevalence of cardiovascular diseases, the need for effective ambulatory and patient monitoring, and recent advancements in sensor technologies. The development of CNIBP sensors requires a framework for reliable and reproducible evaluation. In early stages of development, it is crucial to have a known reference waveform to validate the signal measured by the developed sensor. However, repeatability cannot be guaranteed when working with human subjects, as their waveform can change significantly in short time periods. These slight changes make waveform comparison difficult. Additionally, finding subjects with an abnormal blood pressure waveform can be difficult, which may lead to false conclusions when evaluating the sensor’s capabilities.

For the above-mentioned reasons, a CNIBP simulator would be a valuable tool to support early-stage validation and evaluation of CNIBP sensors. Such devices have already been presented in the literature, including solutions using pre-recorded or simulated waveforms, as well as devices that simulate the physical properties of blood vessels. Most of these systems involve high costs. A low-cost, reliable simulator addresses this issue and supports early-stage sensor development.

A common approach to blood pressure (BP) signal simulation involves building an artificial wrist containing a model of the radial artery that is actuated pneumatically or hydraulically. Validating MediWatch prototypes (MW-1 and MW-2 prototypes, HealthSTATS International; Singapore), Ng et al. [1] used a pneumatic pressure-pulse generator driven by waveform simulation software that actuated a latex diaphragm placed on the artificial wrist’s contact zone. The contact zone of the artificial wrist can also be actuated with a linear motor, such as Heo et al.’s device [2], designed for training oriental medicine practitioners and sensor validation.

Yang et al. developed a cam-based radial pulsation simulator [3] that was subsequently used by Jun et al. [4] for the evaluation of a robotic pulse sensor. This simulator reproduces a single period of a representative waveform, obtained by averaging 40 recorded waveform periods. In their implementation, a rotating cam is connected to a piston, and the pressure wave is transduced pneumatically to an artificial radial artery embedded in a silicone wrist surface. The authors measured a 3.2% phase delay and reported a coefficient of variation (CV) of CV=0.23% and CV=0.82% for heart rate and pulse pressure, respectively. This solution was subsequently developed into a universal cam generating a single representative cardiac cycle of four age groups, with a transitional profile in between (Kim et al. [5]).

Hsieh et al. used a similar cam-based approach for the validation of their tactile pulse recorder [6]. However, their single cardiac cycle cam was fabricated using 3D printing, and the cam surface is in direct contact with the sensing tip of the pulse sensor. This results in a low-cost and simple design, which, however, lacks configurability—an aspect that was beyond the scope of their study. They introduced a pulse tactile player device in the same paper, with an actuating element consisting of piezoelectric benders. This device is intended to train traditional Chinese medicine (TCM) physicians in manual pulse taking in a consistent and repeatable manner. These benders are capable of generating a force of 0.5N with a maximum displacement of 1mm.

McLellan et al. designed a BP pattern simulator [7] that is intended for training practitioners of pulse diagnostics. Therefore, the device is designed to simulate manual pulse sensing using three fingers, with a microprocessor-controlled solenoid assigned to each finger. The device also measures the gripping force with a pressure sensor placed at the opposite side, under the thumb. They implemented a combination of three waveforms and four rhythms, although no evaluation results are reported in the article.

Another approach is to closely replicate the characteristics of the human cardiovascular system. Yang et al.’s simulator [8] consists of modules that simulate the heart and the valves, the aorta with a bifurcation to generate reflected waves, and peripheral resistance, including the wrist with the radial artery. This complex system is also capable of generating age-dependent BP waveforms by simulating arterial stiffness.

In this manuscript, a simulator design that plays back a predefined signal using a mechanical transmission system and a validation framework in Python is introduced. In the Section 2, the implementation details and the method of evaluation are described. In the Section 3, the ground truth signal is compared with the one measurable on the device, and in the Section 4 and Section 5, some further improvements are proposed.

## 2. Materials and Methods

### 2.1. Hardware Design

Most of the above-mentioned solutions aim to be widely applicable and simulate not just a signal waveform but also the measurement environment and the physiological behavior. Therefore, these solutions are complex, and their implementation requires advanced manufacturing technologies. The simulator presented here, on the other hand, can be manufactured and assembled with limited technical resources and budget. Its parts in direct contact with the sensor follow a modular design and are easily customizable; therefore, they can be easily adapted to various sensors under development.

Cam-follower-based designs are frequently used in a wide range of industrial applications, notably in valve control and mass production machines, as they provide a reliable and easy-to-design solution for function generation [9]. In the field of cardiovascular signal simulation, Yang [3] used a similar approach; however, they actuated a pneumatic system with the cam. In the presented simulator design, shown in Figure 1, the sensor to be tested is directly actuated by a planar linkage connected to the cam follower.

To ensure affordability in production, a modular design was implemented, where all components are either 3D-printable or standard, widely available parts. The simulator consists of three main components mounted on a DIN rail $R_{h}$, which allows easy assembly and modification and ensures proper alignment of the lever with respect to the cam.

(1) The waveform generator consists of the interchangeable cam *C* rotated by the motor *M*, a 12V DC motor (GW370 worm gear motor, Shenzhen Jinshunlaite Motor Co., Ltd.; Shenzhen, China) at a nominal speed of 10 rpm (revolutions per minute), approximating the normal pulse rate. The preference for a simple DC motor over a stepper motor is due to reduced cost and complexity. A DC power supply unit EB2025T (Thurlby-Thandar Instruments Ltd.; Huntingdon, UK) was used. A 2 mm thick Poly(dimethylsiloxane) (PDMS) sheet was placed between the motor and the motor base to reduce high-frequency vibration noise. Similarly, PDMS or rubber dampers can be placed between the simulator board and the desk on which it is mounted.

(2) The scaling mechanism achieves a realistic, physiologically relevant amplitude. This mechanism includes lever *L* supported by fulcrum $J_{1}$ and cam follower $J_{2}$, vertical rail $R_{v}$, and translating follower *T*, which ensures that the sensor is actuated only along the *z* axis.

(3) The sensor holder unit *H* is customizable to accommodate sensors of different shapes and sizes. *T* has an interchangeable contact head at *P* to provide an optimal surface for the measurements. The plate surrounding the contact head is also easy to replace to allow additional components like artificial skin or structures that better mimic the geometry of the human wrist. The sensor’s normal vector can be adjusted by three screws through the sensor holder. The baseline pressure can be set by moving the module along its vertical rail $R_{v}$, which adjusts the sensor holder’s height $h_{S}$.

The scaling factor, both in terms of force and displacement, is the ratio of the distances $r_{1}$ and $r_{2}$, where $r_{1}$ is the distance of $J_{2}$, $r_{2}$ is the distance of $J_{3}$, measured from the fulcrum $J_{1}$: $\frac{F_{2}}{F_{1}}=\frac{r_{2}}{r_{1}}$. Therefore, at a given rotation angle θ, the vertical displacement of the sensor contact point *P* is $\frac{r_{2}}{r_{1}}h\left(\theta \right)$, where *h* is the amplitude of the pitch curve of the cam. Depending on configuration, either a class 1 or a class 2 lever can be assembled from the same mechanical parts (Figure 2). This choice does not affect the output waveform; however, in the case of signals with steep rising edges, a cam profile with an inverted signal and class 1 lever can be advantageous.

### 2.2. Cam Design

The cam plays a critical role as it generates the blood pressure waveform. To construct it, noninvasive arterial blood pressure waveforms were used from the PhysioNet [10] dataset titled ”Autonomic Aging: A dataset to quantify changes of cardiovascular autonomic function during healthy aging” [11] (hereafter referred to as AAC). From the 15 age groups presented in the dataset, eight visibly distinct waveforms were selected. All of these signals were measured with Task Force Monitor (TFM) or CNAP 500, CNSystems; MP150, BIOPAC Systems at an $f_{s}=1000\;Hz$ sampling frequency. Details of the selected signals are presented in Table 1 and Figure 3.

First, for noise reduction in the selected signals, a low-pass Butterworth filter was applied with cutoff frequency $f_{max}=30\;Hz$. Subsequently, a characteristic point detection algorithm introduced in Section 2.8 was used to assist in baseline wander (BW) removal via cubic spline interpolation. Then, N=6 consecutive cardiac cycles were chosen. This waveform was normalized between 0 and $y_{max}$. The pitch curve coordinates (x,y) of the cam were then calculated by adding the signal amplitude h=h(θ) to the cam’s baseline radius $r_{C}$, followed by converting to Cartesian coordinates:(1)$$\left[\begin{array}{c} x\\ y \end{array}\right]=\left[\begin{array}{cc} \cos\theta & -\sin\theta \\ \sin\theta & \cos\theta \end{array}\right]\left[\begin{array}{c} h_{0}+h\\ \epsilon \end{array}\right],$$
where $h_{0}=\sqrt{r_{C}^{2}-\epsilon^{2}}$. The baseline radius was set to $r_{C}=30\;\mathrm{mm}$ to fit within the working area of the Sonic Mini 3D printer. Finally, the actual cam profile was calculated using the equations from [12](2)$$\left\{\begin{array}{cc} x_{P} & =x-r_{f}\;\cdot \;\frac{dy/d\theta }{\sqrt{{\left(dx/d\theta \right)}^{2}+{\left(dy/d\theta \right)}^{2}}}\\ y_{P} & =y+r_{f}\;\cdot \;\frac{dx/d\theta }{\sqrt{{\left(dx/d\theta \right)}^{2}+{\left(dy/d\theta \right)}^{2}}} \end{array}\right.,$$
where $\left(x_{P},y_{P}\right)$ are the coordinates of the final cam profile and θ is the rotation angle. These equations take into account the radius $r_{f}$ of the cam follower pin at $J_{2}$. The constraint $r_{f}\ll \kappa^{-1}$ must hold, where κ is the curvature of the cam profile, defined as(3)$$\kappa =\frac{{x_{P}}^{\prime }{y_{P}}^{\prime \prime }-{y_{P}}^{\prime }{x_{P}}^{\prime \prime }}{{\left({x_{P}}^{\prime 2}+{y_{P}}^{\prime 2}\right)}^{3/2}},$$
where the notations $a^{\prime }\equiv da/d\theta$ and $a^{\prime \prime }\equiv d^{2}a/d\theta^{2}$. The minimum radius of curvature for the cams used is $\kappa^{-1}=1.28\;\mathrm{mm}$. Thus, $r_{f}=0.5\;\mathrm{mm}$ is a good choice. Measurements were conducted with $y_{max}\in \left\{0.75;1;1.5\right\}\mathrm{mm}$. Smaller amplitude has the advantage of smaller pressure angle ϕ and smaller lever length $r_{1}$, but may reach the accuracy limitations of 3D printing.

A Python software package was developed that automates the aforementioned steps of designing a cam based on an input waveform. It also adds a cylindrical side protrusion (baseline rim) near the baseline of the signal ($r_{rim}=r-1\;\mathrm{mm}$), that helps detect printing distortions and positioning errors. The cam follower at $J_{2}$ has two low-friction stainless steel pins, one following the signal contour and one for the baseline rim. The software uses the OpenSCAD 2021.01 code-based Computer Aided Design (CAD) tool [13] to generate a 3D-printable STL file from the auto-generated SCAD code.

Two 3D printing technologies were tested: (1) a Craftbot printer (CraftUnique Kft.; Budapest, Hungary) with polylactic acid (PLA) filament and (2) a Sonic Mini 8k printer (Phrozen Technology Co., Ltd.; Hsinchu, Taiwan) with Phrozen resin (TR300). Craftbot with PLA filament was used for printing all structural and support parts of the simulator.

### 2.3. Software

Two software packages were developed in the Python programming language. First, the package bpwave [14] provides a general-purpose signal representation data structure, bpwave.Signal, for Arterial Blood Pressure (ABP) signals. It stores the time series, characteristic points, marks (named indices), and measurement metadata.

Second, the cam_bpw_sim package [15] contains algorithms specifically designed to support simulator development. These include cam generation, measurement evaluation, and scripts for reproducing the results presented in this article. The package has a low-level and a high-level Application Programming Interface (API) and a command-line application. The cam_bpw_sim.cam module supports generating printable cams from arbitrary time series, including the preprocessing, transformation, and calculation steps described in Section 2.2. A quality check is included to warn the user if the signal is unsuitable for a cam with the provided parameters (undercutting happens). The cam_bpw_sim.signal module provides extensions to the aforementioned bpwave package for normalization, resampling, denoising, characteristic point detection (see Section 2.8), and baseline wander correction. Interfaces let users implement their own characteristic point detection and baseline correction algorithms as drop-in replacements of our implementations in the evaluation code. The results presented in this article were calculated using the cam_bpw_sim.val module. The validation workflow is implemented using Jupyter notebooks to facilitate easy customization and detailed inspection of intermediary results. The application saves cam models, signals, and measurement files in a structured format, along with the metadata required to reproduce validation results. These are stored in HDF5 and JSON file formats to ensure interoperability with other software platforms, such as MATLAB (The MathWorks, Inc.; Natick, MA, USA).

### 2.4. Configuration

The user can control the output signal without developing new components, using the adjustable joints and the cam. The adjustable parameters include the following:the cam profile, which defines the input waveform h(θ);$r_{1}\in \left[200,600\right]\;\mathrm{mm}$, the distance of $J_{1}$ and $J_{2}$ along the lever *L* (the fulcrum can slide on *L*);$r_{2}\in \left[20,200\right]\;\mathrm{mm}$, the distance of $J_{1}$ and $J_{3}$ along *L* ($R_{v}$ can slide on $R_{h}$);ϵ∈[−5,5]mm, the eccentricity, the horizontal distance of $J_{2}$ and *O*;and $h_{S}\in \left[135,140\right]\;\mathrm{mm}$, the height of sensor along the vertical rail $R_{v}$.

Adjusting these parameters allows fine-tuning of the baseline force applied to the sensor. These parameters are summarized in Table 2. These values can be passed to the software via the MeasEnvironment Python class, which is automatically filled from the measurement setup configuration and cam metadata files. The user can control supply voltage *U* as well in the range of 10–14 V to simulate an approximation of different heart rates with the same waveform.

### 2.5. Sensor for Validation

As a validation device, the OptoForce OMD-20-SE-40N, a 3D tactile force sensor [16] made by OnRobot, Odense, Denmark (formerly OptoForce, Budapest, Hungary) was used. It consists of a silicone rubber hemisphere with translucent fill and a reflective inner surface. The deformation of the hemisphere is measured optically. Infrared light is reflected onto multiple light-sensing elements, and the amount of reflected light changes as the dome deforms. This allows the calculation and recording of a 3D force vector. As shown in [17], this sensor is suitable for measuring human CNIBP waveforms based on the principle of applanation tonometry.

### 2.6. Validation Protocol

The measurement protocol defines a 3 min 20 s long recording using the OptoForce OMD-20-SE-40N sensor as follows:1.With the force sensor lifted, insert the signal cam.2.Set the sampling frequency to $f_{s}=333\;Hz$, lower the sensor, and adjust its tilt to minimize the *x* and *y* components of the measured force vector.3.Start the motor and adjust $r_{2}$ then $h_{S}$ so that realistic signal amplitude and minimum sensor value (baseline amplitude), respectively, can be acquired.4.Rotate the cam to a position between the zero mark (indicated by the < symbol on the printed model) and the dicrotic peak of the previous cardiac cycle.5.Start the measurement and record 20 s of data from the resting sensor. This segment is used to evaluate the sensor’s baseline noise level and baseline wander.6.Start the motor and record 3 min of the simulated signal.7.Stop the recording.8.Stop the motor.

### 2.7. Validation Measurements

The validation process was carried out using the accompanying software package cam_bpw_sim.

First, cam models were generated for the eight signals introduced in Section 2.2 and printed using a 3D resin printer. The simulator was configured with the following parameters: $r_{1}=500\;\mathrm{mm}$, $r_{2}=23\;\mathrm{mm}$, ϵ=0 mm, and supply voltage U=12 V. $r_{2}\geq 20\;\mathrm{mm}$ because of the horizontal dimensions of the fulcrum and *T*. The value of $r_{1}$ was selected to achieve the desired sensor-specific scaling factor, based on preliminary test measurements, targeting a sensor output in the range of 900–1100 units. Setting ϵ=0 simplifies the calculation in Equation (Equation 1), and 12 V is the recommended supply voltage for the GW370 motor. Finally, the sensor’s tilt and baseline pressure were adjusted.

In total, 55 waveform measurements were conducted using simulator hardware version 1.0. For each of the nine cams (see Table 3), four consecutive measurements were performed: three recordings of the BP waveform and one recording of the baseline rim of the cam with systolic peak marking for synchronization. This measurement sequence was repeated twice. Three additional measurements were conducted with waveform AAC27 prior to the durability testing. Each measurement included approximately 30 full cam rotations. The logged sensor data was processed offline.

### 2.8. Preprocessing of the Measured Data

Prior to statistical analysis, the raw sensor data was preprocessed as illustrated in Figure 4. The Euclidean norm of the logged force components ($\vec{x}$, $\vec{y}$, and $\vec{z}$) was computed to obtain the vector magnitude, forming a time series with the nominal sampling frequency $f_{s}=333\;Hz$, assuming uniform sampling intervals. First, the time series was segmented into noise and signal sections by detecting the first onset through thresholding. Characteristic points were then identified using a custom algorithm, cam_bpw_sim.signal.ScipyFindPeaks, which leverages scipy.signal.find_peaks applied to both the signal and its derivative. This process detects key features, including rising edges, onsets, systolic, dicrotic and reflected peaks, if present, as well as the dicrotic notch.

The statistical analysis requires two different transformed versions of the signal. (1) A signal with long-term baseline wander (BW) removed. This is achieved by subtracting a cubic spline fit calculated from the first onset of each full cam rotation (FCR). This type of baseline wander is attributed to the sensor; removing it makes FCRs comparable while preserving cam baseline errors, which are important for quality assessment. (2) A signal with full baseline correction, obtained by subtracting the cubic spline fit calculated from all detected onsets. This transformation eliminates baseline errors originating from the cam, which are caused by printing distortions such as non-uniform scaling and internal stresses, thereby making the waveform comparable to the nominal reference.

The nominal signal was resampled and transformed to match the sampling frequency and the amplitude of the measured signal. Each signal was segmented into FCRs using an algorithm based on cross-correlation with the adjusted nominal signal. Each cam rotation was further divided into cardiac cycles based on the onset points of the nominal signal. At both segmentation levels, sufficiently large margins were added to the signal segments to ensure inclusion of the first and the last onset points, accounting for uncertainties in point detection. These margins were excluded from waveform comparisons and were only in the statistical analysis of characteristic points.

### 2.9. Measurement Data Analysis

This validation system can evaluate the accuracy, precision, sensitivity, robustness, durability, and reliability of a sensor—or, as in this manuscript, these properties of the simulator itself—by comparing its output to that of a validated sensor.

The accuracy of the simulator was assessed using root mean squared error (RMSE), defined as $\mathrm{RMSE}\left(y,y_{0}\right)=\sqrt{\frac{1}{N}\sum_{i=1}^{N}{\left(y_{i}-y_{0_{i}}\right)}^{2}}$, where *N* is the number of samples, $y_{i}$ is a single measured value, $y_{0}$ is the corresponding nominal value. Pearson correlation coefficients were also calculated, both between pairs of fully measured cam rotations and the nominal signal and between pairs of individual measured cardiac cycles and the corresponding nominal cycles. Additionally, the standard deviation (STD) of the characteristic point positions was computed to assess variability. Since RMSE and STD are expressed in the original scale of the measurements, they can be directly compared to the noise levels observed on the unactuated sensor. Pearson correlation coefficients were calculated using scipy.stats.pearsonr. Precision of the system was evaluated by calculating RMSE between pairs of raw cam rotations, as well as between corresponding characteristic points across measurements.

System durability and reliability were assessed through a one-hour measurement using the AAC27 cam, during which signal error was evaluated over time. Long-term baseline wander was estimated by fitting a cubic spline to the first onset point of each FCR. After baseline correction, short segments at both ends of the signal were discarded to avoid boundary effects from the spline fitting. Within a rolling window of N=30 FCRs, both cross-RMSE (between measured rotations) and the RMSE against the nominal signal were computed.

Three additional measurements were performed to evaluate the effect of different supply voltages.

## 3. Results

The unit of the RMSE in the following sections is the sensor output values as measured by the OptoForce sensor, unless otherwise specified.

First, the accuracy of the simulated waveforms was determined in terms of RMSE and Pearson’s correlation coefficient ρ, by comparing entire waveforms of baseline-corrected full cam rotations to the corresponding nominal waveform. The resulting statistics are summarized in Table 4. The relative RMSE (${\mathrm{RMSE}}_{rel}$) ranged from 0.0194±0.0044 to 0.0274±0.0063, while the corresponding Pearson correlation ($\rho_{rel}$) ranged from 0.9984±0.0013 to 0.9939±0.0018.

Second, the characteristic points of the measured and nominal signals were compared in terms of both timing and amplitude. It is important to note that this evaluation may be influenced by the accuracy of the point detection algorithm, as characteristic points can be ambiguous in regions with flat peaks or shoulder-like features.

Precision was assessed by cross-comparing full cam rotations, with the long-term baseline wander removed (see Figure 5). The RMSE value ranged from 0.0153±0.0071 to 0.0213±0.0116, while Pearson’s correlation coefficient ρ ranged from 0.9985±0.0016 to 0.9959±0.0057. These results are summarized in Table 5.

To evaluate sensitivity, the dimensional accuracy of the 3D printers was assessed using the baseline rim feature of the printed cams. Measurements of this cylindrical feature at the onset positions yielded diameters of 58.02±0.03 mm for resin cams and 57.88±0.06 mm for PLA cams, compared to the nominal 58mm. In this setup, concave regions of the waveform must have a radius of curvature greater than 0.5mm to avoid undercutting. For rising edges approaching vertical slopes, an inverted cam design is recommended with a class 1 lever configuration.

To assess reliability, the median cross-RMSE over the full measurement period was approximately 1.6 units of amplitude. The median cross-RMSE within a rolling window of 30 FCRs varied between 1.5 and 2 units, maintaining a stable level over time (see Figure 6, which indicates that the system’s precision does not degrade significantly during measurement. In terms of accuracy, the rolling RMSE relative to the nominal signal remained at a median level of approximately 3.5 units until around 2500 s, after which the error began to increase. This increase can be attributed to the slowly decreasing amplitude over time, likely caused by a sensor-specific issue, and to a reduction in motor speed due to the power supply limitations.

The proposed system can simulate BP waveforms with different speeds, controlled by the input voltage supplied to the DC motor. Measurements using the selected AAC27 signal showed that changing the supply voltage in the range of 10–14 V had no significant effect on precision. However, accuracy was highest at the nominal voltage of 12V (see Table 6).

Finally, the impact of lever configuration on signal generation was evaluated. To isolate the effect of configuration alone, $r_{1}$ and $r_{2}$ were kept constant. Two measurements were compared: one using a class 2 configuration using the cam AAC27_rf50_Phr1 and one using a class 1 configuration using the inverted cam AAC27i_Phr2. The amplitudes of pressure (125.8 vs. 122.5), precision (1.9283±0.6495 vs. 2.0123±0.4673), and accuracy (2.5609±0.2946 vs. 3.0925±0.1388) were comparable. However, the inverted cam appeared more sensitive to printing artifacts near waveform peaks, likely due to additive surface distortions. FCR-wise, the cross RMSE between the class 1 and class 2 time series was 7.5293±0.4988 in sensor output units. See visual comparison in Figure 7.

## 4. Discussion

The simulator presented here is intended to support the early-stage development of CNIBP sensors utilizing applanation tonometry. The simulator produces vertical motion that follows a waveform defined by the cam profile. The mechanical components in direct contact with the sensor under evaluation are designed to be easy to customize; in particular, the pressure tip can be modified to present different contact surfaces—such as point-like, planar, or cylindrical—mimicking the shape of the pulsating surface of the radial artery. An elastic coating, such as PDMS, may also be applied to the pressure tip to replicate the damping effect of soft tissues between the arterial wall and the sensor. The choice of contact surface depends on the sensor geometry and the specific testing conditions. It is also important that simulated measurements yield sensor outputs within the range expected in human subjects. To accommodate varying different sensitivities, the simulator enables easy adjustment of baseline and pulse amplitudes via the parameters $h_{S}$ and $r_{2}$, respectively. The cam surface is also highly customizable: besides representing baseline-corrected BP waveforms, baseline wander can also be intentionally introduced. In such cases, the baseline wander correction step should be omitted during evaluation. The number of cardiac cycles per full cam rotation depends on two factors: the motor’s rotation speed and the fabrication feasibility of the cam, specifically, constraints related to pressure angle and curvature.

The simulator was validated using the OMD-20-SE-40N 3D force sensor, which has been previously shown to provide high-precision applanation tonometric measurement of continuous blood pressure waveforms on the radial artery [17]. However, known limitations of the sensor affect the measurement results presented here, namely measurement noise and long-term baseline decrease. As a result, the error metrics reported in this study reflect both the printing inaccuracies and the measurement noise inherent to the OptoForce sensor.

The evaluation results indicate that the simulator is capable of reproducing various types of pre-recorded waveforms with sufficient precision and accuracy for both long (45 min) and short (2–5 min) timeframes, fulfilling the requirements for monitoring and diagnostic applications. On a static surface, the average noise amplitude measured by the force sensor ranges between three and six sensor output units. Therefore, the reported precision and accuracy errors are comparable in magnitude to the inherent sensor noise.

A major limitation of the presented simulator design is that the fidelity of the simulated waveform strongly depends on the quality and accuracy of 3D printing used to fabricate the cams. Cams printed from PLA filament tend to be durable and have smooth surfaces, but the use of belt-driven printer heads introduces geometric distortions in the printed part. However, this distortion can be approximated and partially compensated by applying a non-uniform scaling, derived from the measurement of a calibration specimen printed prior to the cam. A corresponding calibration model is provided in [18]. Stereolithography (photopolymer resin printing), in contrast, offers high geometric accuracy but tends to produce less smooth surfaces on features perpendicular to the XY plane, such as the cam profile, due to pixelation, rounding artifacts, and post-processing effects. For both tested printing technologies, accuracy was influenced by the complexity and feature-richness of the desired waveform. It was observed that in resin-printed cams, waveform segments between the dicrotic peak and the subsequent onset were more susceptible to printing noise. Beyond printability, if the target waveform violates the curvature constraint $\kappa^{-1}<r_{f}$, a feasible solution is to reduce the number of cardiac cycles per cam rotation and increase the rotation speed. cam_bpw_sim includes tools to identify problematic cam regions, which can then be addressed by adjusting printing parameters or through manual post-processing. For direct cam quality assessment, currently, the following method can be used. A marker was placed on the cam follower head ($J_{2}$) and a video was recorded with a high-frame-rate camera (Google Pixel 5 with external Hama macro lens, via the OpenCamera app). Similarly to the aforementioned measurement protocol, the recording began at the marked onset of the cam. Ten rotations were included at a frame rate of approximately 240 Hz. The marker position was tracked offline, and an averaged displacement profile was extracted and compared to the nominal waveform. The best-performing cam, printed with Phrozen TR300 resin, yielded a 3.11% NRMSE. In comparison, PLA filament with compensation of uneven scaling showed 5.7% and Phrozen Nylon-Green resin had 6.62%. Note that these errors reflect both eccentricity and profile deviations.

The fabrication complexity and cost of the proposed simulator are significantly lower than those of the existing solutions introduced in Section 1, primarily because previous designs rely on advanced and costly control hardware and materials, whereas the present system can be assembled using standard components and 3D-printed parts with basic tools. These characteristics make the proposed design a cost-effective option for smaller laboratories or research groups with limited resources. In comparison with the existing 3D-printed cam-based simulators (Hsieh et al. [6]), the proposed system generates waveforms with both realistic and adjustable amplitude. Furthermore, it extends prior work [3,5,6] by incorporating a cam profile that represents multiple cardiac cycles, rather than a single cardiac cycle.

After assembly, it is recommended to calibrate both the system and the cams using a gauge capable of continuous force or displacement measurement. Then, cam_bpw_sim can be used for evaluation.

## 5. Conclusions

A hardware–software system was successfully implemented to provide a validation environment to aid the development of continuous noninvasive arterial blood pressure sensors. The simulator is easily configurable to adapt sensors with different measurement ranges and precision, enabling test measurements with amplitudes comparable to those observed in human subjects. Unlike existing solutions [3,5,6], the presented system includes a software package that enables developers to generate cams for arbitrary waveforms via rapid prototyping and to evaluate both the simulator and the pressure sensors of their choice. In line with the Open Science initiative, the entire design is open-source and available at [15,18,19] under an MIT license. The system was developed entirely using free and open-source software.

The simulator was validated using nine cams designed to represent human ABP waveforms across different age groups and physiological characteristics. Measurements with the OptoForce OMD-20-SE-40N sensor demonstrated that the target waveforms can be reproduced with an error comparable to the measurement error of the sensor. Precision error ranged from 1.53 ± 0.71% to 2.13 ± 1.16%, while accuracy values ranged from 97.94 ± 0.44% to 97.72 ± 0.63%. In terms of coefficient of variation (CV), the simulator presented here demonstrated a repeatability ranging from CV=0.5% to CV=0.55%, depending on the cam used. For heart rate repeatability, CV values were between 0.44% and 1.04%. However, this quantity strongly depends on the characteristic point detection algorithm or on variability in the DC motor’s performance.

Consequently, a simple, easily reproducible, yet robust blood pressure waveform simulator was developed to support the early-stage development of CNIBP waveform measurement systems based on pressure or force sensors. This simulator is particularly well-suited for testing force sensors that rely on mechanical deformation of elastic materials, as it enables precise control of the pressure head displacement through the cams and the scaling mechanisms.

Future improvements in waveform quality, particularly long-term accuracy, could be achieved by replacing the DC motor with a stepper motor; however, this would increase production costs. Additionally, the introduction of an artificial skin layer, made of Rubosil (a silicone product of Bondex Szilikontechnika Kft.), is planned to simulate the damping, friction, and elasticity of skin and subcutaneous tissues (Figure 8).

## Figures and Tables

**Figure 1 sensors-25-05173-f001:**
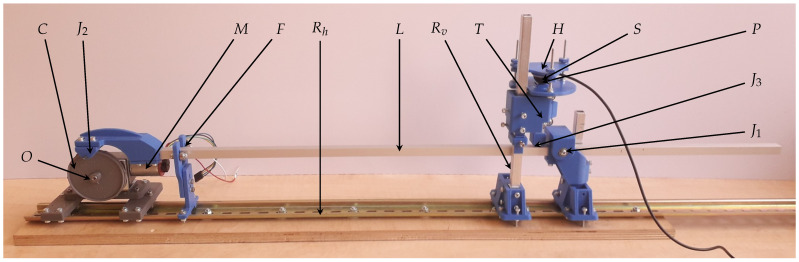
Parts of the simulator: an exchangeable cam *C* with center *O* rotated by motor *M*; lever *L* between revolute joint $J_{1}$ and the half joint of the cam follower $J_{2}$; a translating follower *T* attached to $J_{3}$ at adjustable horizontal and vertical positions along rails $R_{h}$ and $R_{v}$ in contact with sensor *S* at point *P*; sensor-specific holder *H*; all parts mounted on rail $R_{h}$. Lever *L* is also guided by fork *F*.

**Figure 2 sensors-25-05173-f002:**
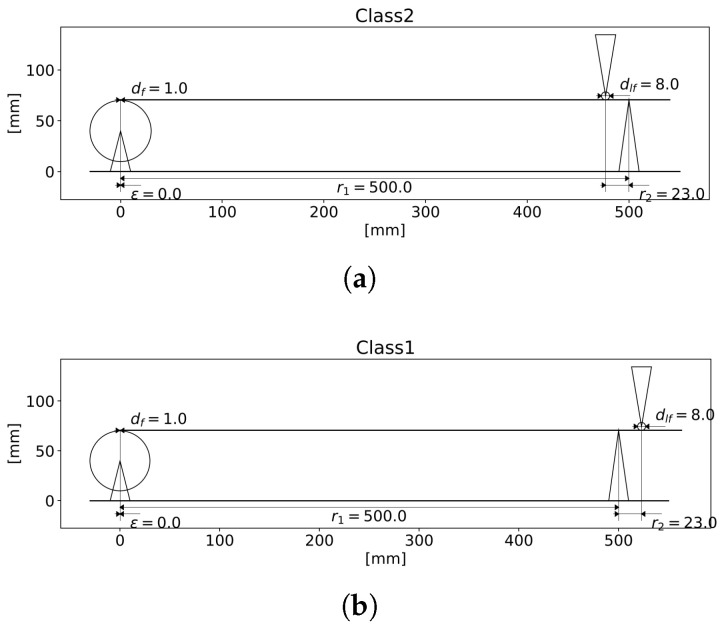
Schematic drawing of the simulator, visualizing a class 2 arrangement (**a**) as generated by MeasEnvironment.simulator.draw(). $d_{f}$ and $d_{lf}$ are the follower diameters, design choices based on cam profile curvature and durability. In the case of a class 1 setup (**b**), the fulcrum is between the cam and the sensor holder.

**Figure 3 sensors-25-05173-f003:**
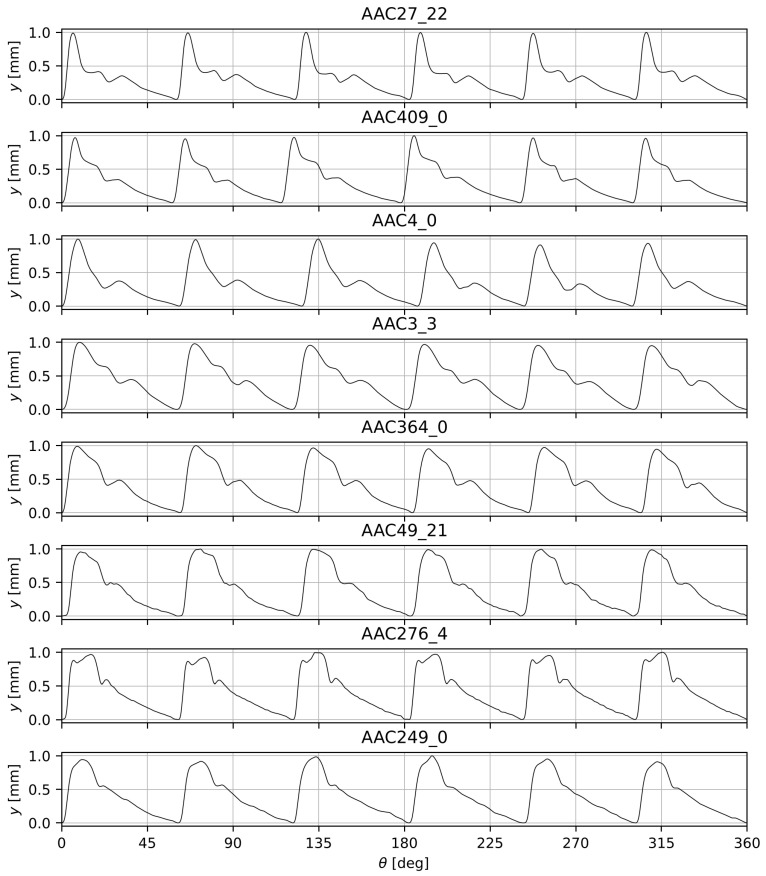
Waveforms of the cams ordered by age group (amplitude vs. rotation angle).

**Figure 4 sensors-25-05173-f004:**
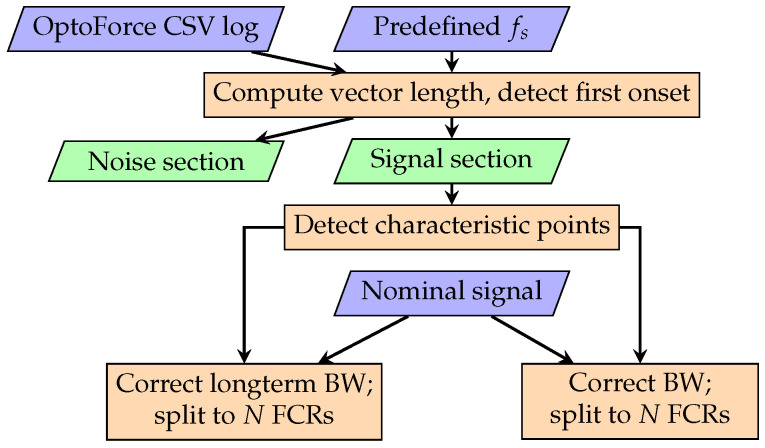
Flowchart of signal preprocessing.

**Figure 5 sensors-25-05173-f005:**
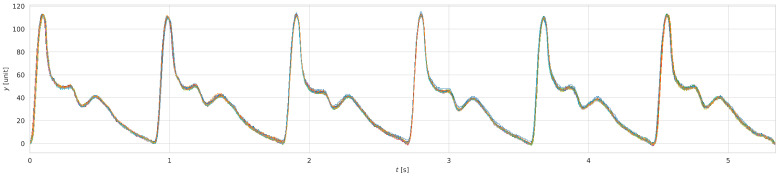
Thirty-two full rotations of the AAC27 cam, aligned using cross-correlation after removal of long-term BW. Consecutive cycles are shown in different colors.

**Figure 6 sensors-25-05173-f006:**
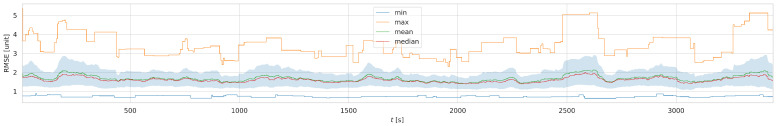
Rolling cross-RMSE of the long-term measurement with respect to time.

**Figure 7 sensors-25-05173-f007:**
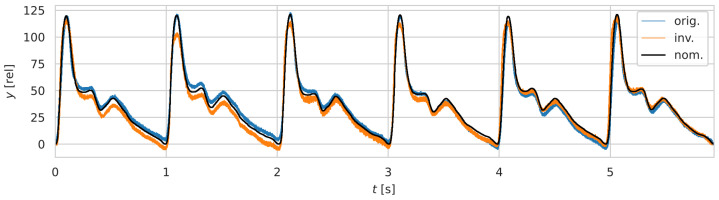
Comparison of stacked FCRs measured with a class 2 configuration (”orig.”), a class 1 configuration (”inv.”), and the nominal signal (”nom.”).

**Figure 8 sensors-25-05173-f008:**
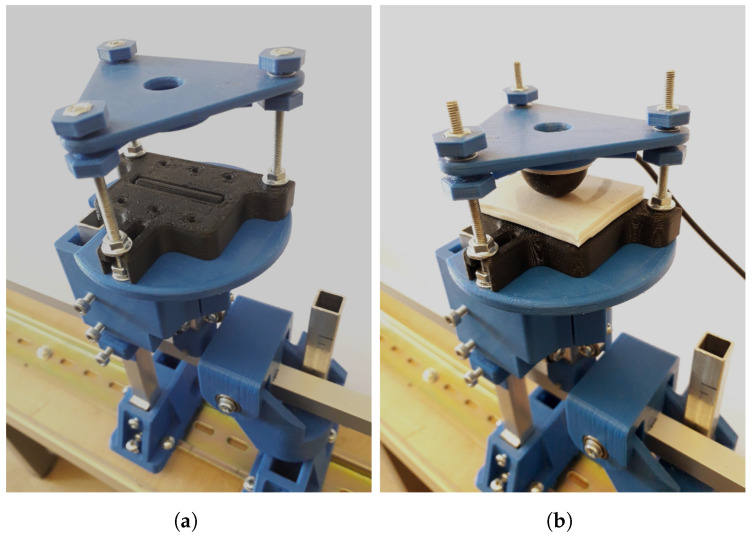
The extension module for simulating the effects of skin on the measurement. The elongated pulsating element (**a**) can be covered with layers of silicone skin (**b**).

**Table 1 sensors-25-05173-t001:** Signal sample selection from [11].

Signal ID	Record ID	Age Group	From *i*th Onset
AAC27_22	0027	18–19	22
AAC409_0	0409	25–29	0
AAC4_0	0004	30–34	0
AAC3_3	0003	45–49	3
AAC364_0	0364	50–54	0
AAC49_21	0049	60–64	21
AAC276_4	0276	75–79	4
AAC249_0	0249	80–84	0

**Table 2 sensors-25-05173-t002:** Configuration of the system. cam_bpw_sim versions are 0.1.0/0.1.1. Supply voltage is included in the class MeasWithMeta that extends MeasEnvironment.

Notation	cambpwsim.meas.MeasWithMeta	Description	Range/Value
h(θ)	nominal	Cam profile (pitch curve)	$y_{max}\leq 1.5\;\mathrm{mm}$
(none)	simulator.lever_class	Lever class I. or II.	1 or 2
$r_{1}$	simulator.lever_r1	The distance of $J_{1}$ and $J_{2}$ along *L* (the fulcrum can slide on *L*)	200–600 mm
$r_{2}$	simulator.lever_r2	The distance of $J_{1}$ and $J_{3}$ along *L* ($R_{v}$ can slide on $R_{h}$)	20–200 mm
ϵ	simulator.fulcrum_x - lever_r1	Eccentricity, i. e. the horizontal distance of $J_{2}$ and *O*	|ϵ|≤5 mm
$d_{f}=2r_{f}$	simulator.cam_follower_d	Diameter of the cam follower	Currently 1mm
$h_{f}$	simulator.cam_follower_h	Vertical distance between the cam follower and the fulcrum shaft	Currently 0
$d_{lf}$	simulator.lever_follower_d	Diameter of the rolling head of *T*	Currently 8mm
$h_{S}$	(not used directly)	The height of sensor along the vertical rail $R_{v}$	135–140 mm
*U*	meas_params.u	Supply voltage	10–14 V
$r_{C}$	cam_params.r	Baseline radius of the cam	Currently 30mm
$r_{rim}$	cam_params.r_rim	Radius of the baseline rim	Currently 29mm
$d_{O}$	cam_params.d_shaft	Diameter of the motor shaft	6mm for GW370
(none)	cam_params.invert	Invert pitch curve	boolean

**Table 3 sensors-25-05173-t003:** Summary of cam and measurement properties. $r_{f}$ [mm] refers to the cam follower radius for which the cam was designed. Meas. ampl. [sensor unit] indictes the mean pulse pressure amplitude recorded during measurements, and # meas. denotes the number of measurements performed using the same cam. The pitch curve amplitude was fixed at $y_{max}=1\;mm$ for all cams. All were printed using Phrozen TR300 resin. Cam AAC27i_Phr2 is an inverted version of AAC27_rf50_Phr1, specifically designed for class 1 lever configuration.

Cam	Signal	$r_{f}$	Meas. Ampl.	# Meas.
AAC249_Phr2	AAC249_0	0.0	121.12	6
AAC276_Phr2	AAC276_4	0.0	124.50	6
AAC27_rf50_Phr1	AAC27_22	0.5	120.42	6
AAC27i_Phr2	AAC27_22	0.5	126.39	6
AAC364_Phr1	AAC364_0	0.5	130.76	6
AAC3_Phr1	AAC3_3	0.0	124.03	6
AAC409_Phr1	AAC409_0	0.5	122.85	6
AAC49_Phr1	AAC49_21	0.5	132.00	6
AAC4_Phr1	AAC4_0	0.0	122.51	6

**Table 4 sensors-25-05173-t004:** Accuracy of the simulated waveforms compared to the nominal signal. #Comp indicates the total number of comparisons, *E* represents the error as defined by RMSE(measuredFCR,nominalwaveform), in original units and $E_{rel}$ denotes the relative error compared to the mean pulse pressure amplitude.

Cam	#Comp	$\bar{E}$	σ(E)	med(E)	$\bar{E_{\mathrm{rel}}}$	$\sigma (E_{\mathrm{rel}})$	$\mathrm{med}(E_{\mathrm{rel}})$
AAC409_Phr1	168	2.3784	0.5350	2.1962	0.0194	0.0044	0.0179
AAC249_Phr2	181	2.4897	0.3988	2.3987	0.0206	0.0033	0.0201
AAC49_Phr1	167	2.8349	0.6133	2.6603	0.0215	0.0047	0.0203
AAC4_Phr1	181	2.7831	0.4078	2.6588	0.0228	0.0031	0.0220
AAC364_Phr1	171	3.0110	0.7778	2.8297	0.0230	0.0059	0.0217
AAC27_rf50_Phr1	180	2.9150	0.5719	2.7710	0.0243	0.0043	0.0235
AAC3_Phr1	177	3.0547	1.3411	2.5577	0.0247	0.0112	0.0201
AAC27i_Phr2	168	3.2443	0.4224	3.1291	0.0257	0.0033	0.0250
AAC276_Phr2	176	3.4223	0.8719	3.1456	0.0274	0.0063	0.0253

**Table 5 sensors-25-05173-t005:** Precision measurements: cross comparison of full cam rotations. #Comp denotes the total number of pairwise comparisons (excluding self-comparisons). *E* represents the error (RMSE), in sensor output units and $E_{rel}$ is the relative error compared to the mean pulse pressure amplitude.

Cam	#Comp	$\bar{E}$	σ(E)	med(E)	$\bar{E_{\mathrm{rel}}}$	$\sigma (E_{\mathrm{rel}})$	$\mathrm{med}(E_{\mathrm{rel}})$
AAC4_Phr1	2654	1.8738	0.9060	1.5672	0.0153	0.0071	0.0129
AAC409_Phr1	2269	1.9543	0.8315	1.7116	0.0159	0.0068	0.0139
AAC249_Phr2	2650	1.9404	0.8380	1.7102	0.0160	0.0067	0.0142
AAC3_Phr1	2562	2.1520	1.2036	1.7115	0.0173	0.0092	0.0140
AAC49_Phr1	2242	2.4679	1.3009	2.0093	0.0187	0.0099	0.0152
AAC27_rf50_Phr1	2623	2.2697	1.2290	1.9015	0.0188	0.0095	0.0161
AAC276_Phr2	2503	2.3824	1.5502	1.8287	0.0190	0.0118	0.0149
AAC27i_Phr2	2269	2.5409	1.0033	2.2937	0.0201	0.0078	0.0182
AAC364_Phr1	2352	2.7840	1.5171	2.3596	0.0213	0.0116	0.0181

**Table 6 sensors-25-05173-t006:** Effect of input voltage on accuracy. #Comp indicates the total number of pairwise comparisons (excluding self-comparisons). *E* represents the error (RMSE) in sensor output units and $E_{rel}$ is the relative error compared to the mean pulse pressure amplitude of the corresponding signal.

*U*	#Comp	$\bar{E}$	σ(E)	med(E)	$\bar{E_{\mathrm{rel}}}$	$\sigma (E_{\mathrm{rel}})$	$\mathrm{med}(E_{\mathrm{rel}})$
10.0	75	3.1973	0.4514	3.2091	0.0270	0.0038	0.0271
11.0	88	3.0556	0.5457	2.8793	0.0259	0.0046	0.0244
12.0	97	2.9989	0.4428	2.8660	0.0255	0.0038	0.0244
13.0	107	3.1443	0.3849	3.0756	0.0269	0.0033	0.0263
14.0	112	3.2502	0.4545	3.1846	0.0278	0.0039	0.0272

## Data Availability

The original data presented in the study are openly available in GitHub at https://github.com/repat8/cam-bpw-sim-pub (version 1.0.0) (accessed on 17 August 2025).

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
