# Peer review of "An Open Source Validation System for Continuous Arterial Blood Pressure Measuring Sensors"

_sensors, 2025, doi:10.3390/s25165173_

Round 1

Reviewer 1 Report

Comments and Suggestions for Authors
  1. The authors mention that the accuracy of the simulated waveform is highly dependent on the quality of 3D printing, especially for sharp waveform features. Could the authors quantify the printing-induced waveform distortion across different cam materials and printing resolutions, and clarify whether any compensation mechanism (e.g., post-processing or error calibration) is implemented?
  2. The RMSE values presented for waveform accuracy and precision combine errors from multiple sources, including the simulator mechanics, cam printing, and the OptoForce sensor itself. Have the authors conducted experiments or control tests to isolate and quantify the contribution of each source of error?
  3. The peak detection algorithm (ScipyFindPeaksDetector) is used to locate characteristic points such as systolic and dicrotic peaks. Given the presence of noise and waveform variability, how robust is this method? Have the authors validated its detection accuracy against manually annotated ground truth?
  4. The simulator claims to support various CNIBP sensor designs, but validation was performed only using the OptoForce OMD-20-SE-40N sensor. Can the authors provide evidence or discussion on how well the system performs with other sensor types, especially those with different contact mechanics (e.g., piezoresistive or capacitive sensors)?

Reviewer 2 Report

Comments and Suggestions for Authors

I have the following questions about this paper:

  1. The author may explain how the study takes into consideration the limited supply of aberrant samples and the inherent variability in human blood pressure waveforms.
  2. Could the author elaborate on the drawbacks of Hsieh et al.'s approach that their own solution gets around? Furthermore, what impact does the increased configurability have on the experimental design, particularly with regard to waveform reproducibility and sensor calibration?
  3. How does this inexpensive, modular system perform in terms of repeatability, accuracy, and resilience across various sensor technologies in comparison to other current simulators that demand more sophisticated resources?
  4. The author may offer evidence that the system's high-frequency vibration noise is considerably reduced when PDMS is used. In the absence of frequency-domain analysis or experimental data, how can the efficacy of the damping method be confirmed?
  5. The justification for segmenting the measured data using cross-correlation with the nominal signal may be provided by the author. In the event of noise or signal distortion, how can this technique guarantee precise identification of individual FCRs and cardiac cycles?
  6. Could the author elaborate on the performance comparison between the simulator and a validated sensor? What factors lead to a sensor being "validated," and how is that status used to evaluate the simulator's features?
  7. In comparison to other well-known algorithms, how well does the point identification algorithm employed in this work handle ambiguous waveform features such as inflection points or flat peaks?
  8. The author may include more details on how the application of elastic coatings, such as PDMS, affects the pressure signal quantitatively. Did any studies compare the effects of coated and uncoated tips on signal distortion, delay, or damping?
  9. In terms of maintaining waveform integrity, especially in areas that are sensitive to curvature and surface smoothness, how does resin printing stack up against alternative fabrication techniques like FDM or CNC machining?
  10. Could the author discuss the precise material or structural characteristics that the artificial skin layer is aiming for and how they will be matched to the fat and skin of humans? Will these changes be fixed for particular test cases, or will they be modular?
